

# A molecular phylogeny of *Geotrochus* and *Trochomorpha* species (Gastropoda: Trochomorphidae) in Sabah, Malaysia reveals convergent evolution of shell morphology driven by environmental influences

Zi-Yuan Chang and Thor-Seng Liew

Institute for Tropical Biology and Conservation, Universiti Malaysia Sabah, Kota Kinabalu, Sabah, Malaysia

## ABSTRACT

There are currently eleven *Geotrochus* and four *Trochomorpha* species in Sabah. The primary diagnostic character that separates the two genera is the intensity of sculpture on the shell upper surface. All *Trochomorpha* species have a coarse nodular sculpture while *Geotrochus* species has a non-nodular sculpture or smooth shell. However, it is known that shell characters are often evolutionary labile with high plasticity in response to environmental factors. Hence, identifying the phylogenetic and ecological determinants for the shell characters will shed light on the shell-based taxonomy. This study aims to estimate the phylogenetic relationship between *Geotrochus* and *Trochomorpha* species in Sabah based in two mitochondrial genes (COI, 16S) and one nuclear gene (ITS) and also to examine the influence of temperature, elevation and annual precipitation on the coarseness of shell upper surface sculpture and shell sizes of the species of both genera. Additionally, we also investigated the phylogenetic signal of the shell characters. The phylogenetic analysis showed that *Geotrochus* and *Trochomorpha* species are not reciprocally monophyletic. The phylogenetic signal test suggested that shell size and upper surface sculpture are homoplastic, and these shell traits are strongly influenced by elevation and annual precipitation, particularly at the cloud zone of Mount Kinabalu. The highland species of both genera have a coarser shell surface than lowland species. The shell and aperture width decrease with increasing elevation and annual precipitation. In the view of finding above, the current taxonomy of *Geotrochus* and *Trochmorpha* in this region and elsewhere that based on shell characters need to be revised with sufficient specimens throughout the distribution range of the two genera.

Corresponding author
Thor-Seng Liew,
thorsengliew@gmail.com

## INTRODUCTION

*Geotrochus* and *Trochomorpha* are two land snail genera that with similar shell forms belonging to the family Trochomorphidae (Fig. 1). The species of the two genera are

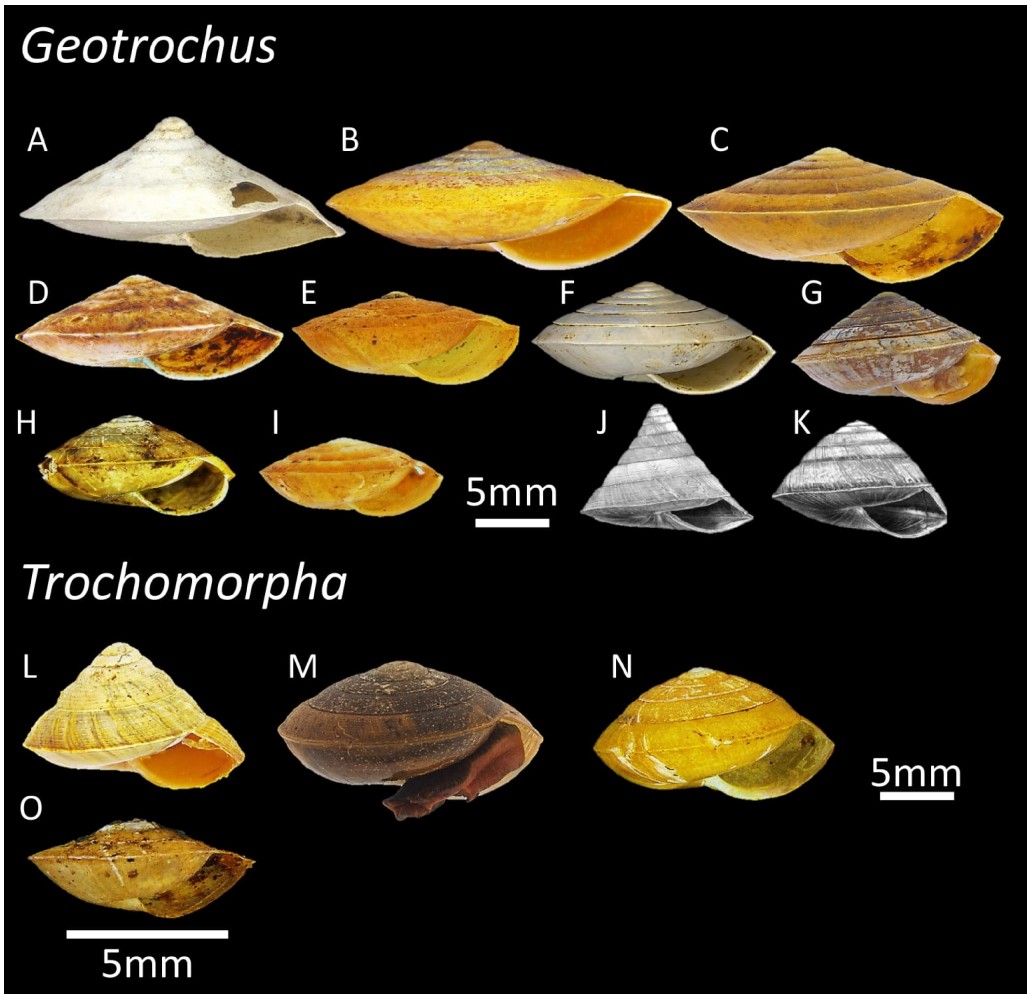

**Figure 1** **The variation of shell forms of 11 *Geotrochus* species and four *Trochomorpha* species in Sabah.** (A) *G. conicoides* (BOL/MOL 2431). (B) *G. paraguensis* (BOL/MOL 13061). (C) *G. kinabaluensis* (BOL/MOL 13020). (D) *G. labuanensis* (BOL/MOL 904). (E) *G. oedobasis* (BOL/MOL 908). (F) *G. subscalaris* (BOL/MOL 2430). (G) *G. meristotrochus* (BOL/MOL 13833). (H) *G. whiteheadi* (BOL/MOL 4110). (I) *G. kitteli* (BOL/MOL 4109). (J) *G. spilokeiria* (image from *Vermeulen, Liew & Schilthuizen, 2015*, CC BY 4.0). (K) *G. scolops* (image from *Vermeulen, Liew & Schilthuizen, 2015*, CC BY 4.0). (L) *T. trachus* (BOL/MOL 2959). (M) *T. haptoderma* (BOL/MOL 6312). (N) *T. rhysa* (BOL/MOL 3986). (O) *T. thelecoryphe* (BOL/MOL 6334).

ground-dwelling snails typically spotted on the understory vegetation and with overlapping distribution ranges in the region of Oceania and Southeast Asia (File S1). A recent revision of both genera reveals a total of eleven *Geotrochus* species and four *Trochomorpha* species in Sabah (*Vermeulen, Liew & Schilthuizen, 2015*). *Trochomorpha* species are endemic to montane forest and subalpine forest between 1,500 m and 3,400 m on Mount Kinabalu and Crocker Range in Sabah, while *Geotrochus* species are widespread in Sabah occur from lowland forest at sea level to highland until 2,400 m (Table 1; *Vermeulen, Liew & Schilthuizen, 2015*).

**Table 1** The number of specimens of *Geotrochus* and *Trochomorpha* species included in the respective phylogenetic analysis and shell morphological analysis in this study.

| Species | Specimens for phylogenetic analysis[a] | Quantitative shell traits[b] | Upper shell sculpture type[c] | | | | Elevational range |
|---|---|---|---|---|---|---|---|
| | | | S1 | S2 | S3 | S4 | |
| *Geotrochus conicoides* (*Metcalfe, 1851*) | NA | NA | NA | NA | NA | NA | 63 m–363 m |
| *Geotrochus kinabaluensis* (*Smith, 1895*) | 2 | 4 | – | – | 3 | 1 | 16 m–2,001 m |
| *Geotrochus kitteli* Vermeulen, Liew & Schilthuizen, 2015 | 1 | 2 | – | 4 | – | – | 1,563 m–2,376 m |
| *Geotrochus labuanensis* (Pfeiffer, 1863) | NA | 16 | – | – | 14 | 2 | 1 m–1,494 m |
| *Geotrochus meristotrochus* Vermeulen, Liew & Schilthuizen, 2015 | 5 | 27 | – | – | 21 | 6 | 9 m–1,680 m |
| *Geotrochus oedobasis* Vermeulen, Liew & Schilthuizen, 2015 | 3 | 6 | – | 1 | 5 | – | 260 m–2,291 m |
| *Geotrochus paraguensis* (*Smith, 1893*) | 8 | 10 | – | – | 9 | 1 | 1 m–756 m |
| *Geotrochus scolops* Vermeulen, Liew & Schilthuizen, 2015 | NA | NA | NA | NA | NA | NA | 718 m |
| *Geotrochus spilokeiria* Vermeulen, Liew & Schilthuizen, 2015 | NA | NA | NA | NA | NA | NA | 1,241 m |
| *Geotrochus subscalaris* Vermeulen, Liew & Schilthuizen, 2015 | NA | 10 | – | – | 5 | 5 | 6 m–988 m |
| *Geotrochus whiteheadi* (*Smith, 1895*) | 1 | 1 | – | – | 1 | – | 827 m–2,080 m |
| *Trochomorpha haptoderma* Vermeulen, Liew & Schilthuizen, 2015 | 8 | 7 | 43 | – | – | – | 2,055 m–3,360 m |
| *Trochomorpha rhysa* Tillier & Bouchet, 1988 | 6 | 5 | 26 | – | – | – | 1,677 m–3,263 m |
| *Trochomorpha thelecoryphe* Vermeulen, Liew & Schilthuizen, 2015 | 1 | 0 | 8 | – | – | – | 1,990 m–2,992 m |
| *Trochomorpha trachus* Vermeulen, Liew & Schilthuizen, 2015 | NA | NA | NA | NA | NA | NA | 1,563 m–1,815 m |

Notes.
[a]The details for the specimens and the accession number of the DNA sequences are available in Table 2.
[b]The number of specimens available for shell quantitative traits measurement, namely, shell height, shell width, aperture height, and aperture width. The full dataset is available in File S8.
[c]The number of specimens available for shell surface sculpture examination, and the variations of the shell sculpture types for *Geotrochus* and *Trochomorpha* species. The full dataset is available in File S8.
NA, No suitable shell was for the DNA data, shell quantitative traits measurement, or shell surface sculpture examination; –, No specimen of the species belongs to the shell surface sculpture type.

Taxonomy of *Geotrochus* and *Trochomorpha* in Sabah has been mainly based on shell and anatomical characters (*Tillier & Bouchet, 1988*; *Vermeulen, Liew & Schilthuizen, 2015*). *Trochomorpha rhysa* is the first species of *Trochomorpha* species described from Sabah (*Tillier & Bouchet, 1988*) from Mount Kinabalu between 3,000 m and 3,500 m. This new species was placed under *Trochomorpha* based on the genitalia and radula characters. After that, more new species of *Trochomorpha* and *Geotrochus* were described solely based on the shell characters (*Vermeulen, Liew & Schilthuizen, 2015*). *Vermeulen, Liew & Schilthuizen (2015)* noted that these species of the two genera have a similar shell, but *Trochomorpha* species have a coarser nodular sculpture on the upper surface of the shell.

Taxonomy of land snails based on anatomy and shell characters are not without its weakness because many of these characters are evolutionary labile (*Pfenninger, Bahl & Streit, 1996*; *Liew, Schilthuizen & Vermeulen, 2009*; *Holznagel, Colgan & Lydeard, 2010*; *Hyman & Ponder, 2010*; *Hirano, Kameda & Chiba, 2014*; *Dowle et al., 2015*; *Köhler & Criscione, 2015*). This open a question to what extent the shell upper surface sculpture is phylogenetically informative in *Geotrochus* and *Trochomorpha* as shell surface sculpture is known to evolve rapidly and in parallel or convergently in response to environmental conditions (*Pfenninger & Magnin, 2001*; *Liew, Schilthuizen & Vermeulen, 2009*). Therefore, it is vital to examine the phylogenetic relationship among *Trochomorpha* and *Geotrochus* species and the influences of habitat climatic factors to clarify the taxonomy of the two genera in Sabah as a way forward to improve the taxonomy of the two genera in Oceania and Southeast Asia in general.

Hence, this study aims to estimate the molecular phylogenetic relationship of selected species of *Geotrochus* and *Trochomorpha* species in Sabah by using two mitochondrial genes (COI and 16S) and one nuclear gene (ITS-1). After that, we examined the association of the shell size and shell upper surface sculptures with several environmental variables in their habitats. Lastly, the phylogenetic signal of the shell characters was tested.

## MATERIALS & METHODS

### Samples

All the eleven *Geotrochus* and four *Trochomorpha* species from Sabah are available in the BORNEENSIS Mollusca collection of Institute of Tropical Biology and Conservation in Universiti Malaysia Sabah. However, not all specimens of the species were suitable for phylogenetic and morphological analysis (Table 1). A total of six *Geotrochus* species, namely, *G. meristotrochus* (*Vermeulen, Liew & Schilthuizen, 2015*), *G. kinabaluensis* (*Smith, 1895*), *G. paraguensis* (*Smith, 1893*), *G. oedobasis* (*Vermeulen, Liew & Schilthuizen, 2015*), *G. kitteli*, (*Vermeulen, Liew & Schilthuizen, 2015*), and *G. whiteheadi* (*Smith, 1895*); and three *Trochomorpha* species, namely, *T. haptoderma* (*Vermeulen, Liew & Schilthuizen, 2015*), *T. rhysa* (*Tillier & Bouchet, 1988*), and *T. thelecoryphe* (*Vermeulen, Liew & Schilthuizen, 2015*) were available phylogenetic analysis. For morphological analysis, a total of 155 specimens of eight *Geotrochus* and three *Trochomorpha* species with intact shells were chosen to obtain quantitative and qualitative measurements. As there is no good quality specimen in the collection for *Trochomorpha trachus* (*Vermeulen, Liew & Schilthuizen, 2015*), *Geotrochus conicoides* (*Metcalfe, 1851*), *Geotrochus spilokeiria* (*Vermeulen, Liew & Schilthuizen, 2015*) and *Geotrochus scolops* (*Vermeulen, Liew & Schilthuizen, 2015*), these species were not included in the present study. Field sampling was approved by the Sabah Parks for Mt.Kinabalu, Tambuyukon, Mahua, Banggi Island and Balambangan Island, and Yayasan Sabah for INIKEA project site, Imbak Canyon and Maliau Basin (Permit: TTS/IP/100-6/2 Jld.7(70), 2018; Maliau Basin TTRP Project No. 228, 2017; and ICCA Expedition 2017).

**Table 2** Species, voucher specimens, location information, and GenBank accession number for the specimens included in the phylogenetic analysis.

| Collection reference number[a] of the voucher specimens (BOR/MOL) | Taxon | Location[b] | Sequence[c] | | |
|---|---|---|---|---|---|
| | | | COI | 16S | ITS-1 |
| 6347 | *Trochomorpha rhysa* | Mount Kinabalu at 3,024 m | MK779474 | MK334188 | MK335437 |
| 6350 | *Trochomorpha rhysa* | Mount Kinabalu at 3,088 m | MK779475 | MK334190 | MK335439 |
| 6353 | *Trochomorpha rhysa* | Mount Kinabalu at 2,944 m | MK779477 | MK334191 | NA |
| 6354 | *Trochomorpha rhysa* | Mount Kinabalu at 2,944 m | MK779479 | NA | MK335440 |
| 6407 | *Trochomorpha rhysa* | Mount Kinabalu at 3,221 m | MK779478 | MK334195 | MK335444 |
| 6411 | *Trochomorpha rhysa* | Mount Kinabalu at 3,119 m | MK779476 | MK334196 | MK335446 |
| 6312 | *Trochomorpha haptoderma* | Mount Kinabalu at 2,775 m | NA | MK334185 | MK335433 |
| 6349 | *Trochomorpha haptoderma* | Mount Kinabalu at 2,896 m | MK779473 | MK334189 | MK335438 |
| 6356 | *Trochomorpha haptoderma* | Mount Kinabalu at 2,800 m | MK779472 | MK334192 | MK335441 |
| 6408 | *Trochomorpha haptoderma* | Mount Kinabalu at 2,484 m | MK779471 | NA | NA |
| 6409 | *Trochomorpha haptoderma* | Mount Kinabalu at 2,526 m | MK779470 | NA | MK335445 |
| 6412 | *Trochomorpha haptoderma* | Mount Kinabalu at 2,500 m | MK779469 | MK334197 | MK335447 |
| 6413 | *Trochomorpha haptoderma* | Mount Kinabalu at 2,404 m | MK779468 | NA | MK335448 |
| 6417 | *Trochomorpha haptoderma* | Mount Kinabalu at 2,896 m | MK779467 | NA | MK335449 |
| 6335 | *Trochomorpha thelecoryphe* | Mount Kinabalu at 2,700 m | MK779480 | NA | MK335434 |
| 6342 | *Geotrochus oedobasis* | Mount Kinabalu at 2,100 m | MK779461 | MK334186 | MK335435 |
| 6404 | *Geotrochus oedobasis* | Mount Kinabalu at 2,200 m | MK811549 | MK334193 | MK335442 |
| 6343 | *Geotrochus oedobasis* | Mount Tambuyukon at 2,080m | MK811548 | NA | NA |
| 6344 | *Geotrochus whiteheadi* | Mount Tambuyukon at 2,080 m | MK811544 | MK334187 | MK335436 |

| Collection reference number[a] of the voucher specimens (BOR/MOL) | Taxon | Location[b] | Sequence[c] | | |
|---|---|---|---|---|---|
| | | | COI | 16S | ITS-1 |
| 6406 | *Geotrochus kitteli* | Mount Kinabalu at 2,300 m | MK779460 | MK334194 | MK335443 |
| 12670 | *Geotrochus kinabaluensis* | Crocker Range, Mahua at 1,200 m | MK811543 | NA | MK335450 |
| 13017 | *Geotrochus kinabaluensis* | Crocker Range, Mahua at 1,200 m | MK811542 | NA | NA |
| 13016 | *Geotrochus meristotrochus* | Tawau, INIKEA site at 200 m | MK811545 | MK334198 | MK335451 |
| 13323 | *Geotrochus meristotrochus* | Imbak Canyon Conservation Area between 400 and 600 m | MK811547 | MK334204 | MK335459 |
| 13325 | *Geotrochus meristotrochus* | Imbak Canyon Conservation Area between 400 and 600 m | MK811546 | MK334205 | MK335460 |
| 13373 | *Geotrochus meristotrochus* | Maliau Basin Conservation Area between 400 and 600 m | NA | NA | MK335461 |
| 13376 | *Geotrochus meristotrochus* | Maliau basin Conservation Area between 400 and 600 m | NA | NA | MK335462 |
| 13061 | *Geotrochus paraguensis* | Kudat, Banggi Island between 50–800 m | MK811550 | MK334199 | MK335452 |
| 13176 | *Geotrochus paraguensis* | Kudat, Banggi Island between 50–800 m | MK811552 | MK334200 | MK335454 |
| 13177 | *Geotrochus paraguensis* | Kudat, Banggi Island between 50–800 m | MK811551 | MK334201 | MK335455 |
| 13223 | *Geotrochus paraguensis* | Kudat, Banggi Island between 50 and 800 m | MK779464 | MK334202 | MK335456 |
| 13224 | *Geotrochus paraguensis* | Kudat, Banggi Island between 50–800 m | MK779465 | NA | MK335457 |
| 13225 | *Geotrochus paraguensis* | Kudat, Banggi Island between 50–800 m | MK779463 | MK334203 | MK335458 |
| 13068 | *Geotrochus paraguensis* | Kudat, Balambangan Island between 20–100 m | MK779462 | NA | MK335453 |
| 13084 | *Geotrochus paraguensis* | Kudat, Balambangan Island between 20–100 m | MK779466 | NA | NA |

**Notes.**
[a] All specimens were deposited at *BORNEENSIS* reference collection at Universiti Malaysia Sabah.
[b] All specimens were collected from the State of Sabah, Malaysia. The elevation of the specimens collected from the habitats was indicated.
NA, The DNA sequence was not available as the amplification of the gene was not successful..

## DNA extraction, amplification and sequencing

Foot muscle with about two mm$^3$ was excised from the preserved land snails using a sterilised scalpel. Genomic DNA was extracted using DNeasy Blood and Tissue Kit (Qiagen Inc., Hilden, Germany) following the standard procedure of the manual. Each of the two mitochondrial genes fragment was amplified by using primer pair LCO1490 and HCO2198 (*Folmer et al., 1994*) with an annealing temperature of 54 °C for COI; and primer pair 16Sbr-L and 16Sbr-H (*Palumbi et al., 1991*) with an annealing temperature of 47 °C for 16S. One nuclear gene fragment (ITS-1) were amplified using the primer pair 5.8c 'silkworm' and 18d' fruitfly' (*Hillis & Dixon, 1991*) with an annealing temperature of 55 °C. The PCR thermal-cycling profile includes initial denaturation at 94 °C for 3 min, followed by 35 cycles of denaturation at 94 °C for 30s, annealing at a locus-specific temperature for each primer for 45s, extension at 72 °C for 1 min and a final extension at 72 °C for 5 min. Positive PCR products were then sent to MyTACG Bioscience Enterprise for sequencing by using the forward and reverse primers that were used during PCR.

## Sequence alignment and molecular phylogenetic reconstruction

The resulting forward and reverse sequences were assembled and aligned in Bioedit 7.2.6 (*Hall, 1999*), and the sequences were deposited in GenBank (Table 2). A total of four DNA sequence data matrix were made—one for each of the markers (16S, ITS, and COI) and one concatenated data matrix of the three markers. For the data matrixes with one marker, each was tested for molecular substitution model by using ModelFinder (*Kalyaanamoorthy et al., 2017*) based on the both AIC and BIC that built into IQ-Tree v.1.6.7 (*Nguyen et al., 2015*; *Trifinopoulos et al., 2016*). However, the COI data matrix was partitioned by codon positions before it was tested for the molecular substitution model.

For concatenated data matrix, it was partitioned by markers and codons (16S, ITS-1, first codon positions of COI, second codon positions of COI, and third codon positions of COI). Each of the partitions was tested for molecular evolution via ModelFinder (*Kalyaanamoorthy et al., 2017*) and partition models (*Chernomor, Von Haeseler & Minh, 2016*) based on the both AIC and BIC that built into IQ-Tree v.1.6.7 (*Nguyen et al., 2015*; *Trifinopoulos et al., 2016*). For all the analyses, we limited the candidate models to the six models that are available in MrBayes analysis, namely, JC, F81, K80, HKY, SYM and GTR. The phylogenetic analyses were performed based on the best partitioning scheme and substitution model for the respective markers and concatenated data matrix (File S2).

Next, we used Bayesian Inference (BI), and Maximum Likelihood (ML) approaches to reconstruct the phylogenetic trees by using MrBayes v3.2.6 (*Huelsenbeck et al., 2001*) and maximum likelihood (ML) method implemented in IQ-Tree v.2.1.1 (*Nguyen et al., 2015*) respectively for the concatenated data matrix and the data matrix for each of the three genes. All analyses were done in the CIPRES Science Gateway portal (*Miller, Pfeiffer & Schwartz, 2010*). The BI analysis was run for 1000000 generations along four chains with sample frequency set to 100 and a burn-in of 2500 (25%) (File S3). The phylogenetic trees generated from the two approaches were then viewed and edited using TreeGraph 2.14 (*Stöver & Müller, 2010*). *Everettia klemmantanica* (Dyakiidae) was selected as an outgroup because this species was the sister taxa of the Trochomorphidae (*Bouchet et al., 2017*).

## Phylogenetic signal analysis

To investigate the influence of phylogeny on the evolution of shell upper surface sculpture and the four quantitative shell traits, the phylogenetic signal of these shell characters were assessed with Pagel's Lambda (*Pagel, 1999*) and Blomberg's K (*Blomberg, Garland Jr & Ives, 2003*). The analysis was performed by using "geiger" package (*Harmon et al., 2008*) and "phytol" package (*Revell, 2012*) in the environment of RStudio 1.1.4 (*RStudio Team, 2015*) following the method of *Phung, Heng & Liew (2017)* (File S4). We used the phylogenetic tree resulted from Maximum Likelihood (ML) but retained only one tip for each taxon, except for *G. paraguensis* which two tips were included—one for each of the two paraphyletic clades. For the qualitative shell trait, all the ten tips with nine species in the phylogenetic tree used for the phylogenetic analysis. However, for the quantitative shell traits, the tips of the phylogenetic tree represented by the juvenile specimen (i.e., *T. thelecoryphe*) were excluded (File S5).

## Shell characters measurement

A total of five primary diagnostic shell characters that were used for delimitation of the species in *Geotrochus* and *Trochomorpha* were measured qualitatively and quantitatively (Fig. 2). The types of shell upper surface sculptures for the adult and subadult specimens with at least three whorls were recorded based on the four categories (S1–S4) of coarseness that are visible at 8× magnification. Sculpture S1—Densely placed, more or less regularly spaced radial riblets and between 11-19 spiral threads that form nodes over the radial sculpture; S2—Raised and distinct radial growth lines and 15 thin spiral threads; S3—Indistinct radial growth lines and inconspicuous riblets and between 6 and 23 thin or very thin spiral threads; and S4—Inconspicuous growth lines and between 4 and 25 low and thin spiral threads. There are a few species exhibit variability in the shell upper surface sculptures. Thus, the specimens of these species can be categorised into two shell upper surface sculpture types.

Also, four quantitative measurements of shell size, namely, shell height (SH), shell width (SW), aperture height (AH) and aperture width (AW) were measured to nearest 0.1 mm from the photograph of the shell apertural view with the aid of Leica Stereo Microscope M205 (Fig. 2). Although there are other shell characters included in the description of each species by *Vermeulen, Liew & Schilthuizen (2015)*, we only included these five primary diagnostic characters due to two reasons. First, the evolutionary and ecological aspects of these selected characters are better known since the review by *Goodfriend (1986)* and second, the other shell characters are species-specific.

## Collection of ecological data

To investigate the correlation between shell size and upper surface sculpture and the environmental variables, we obtained the elevation, annual precipitation and temperature of the location where the specimens were collected. The elevation of the location was extracted from SRTM DEM 30-meter resolution (http://earthexplorer.usgs.gov/), and the annual precipitation and annual average temperature were extracted from global average temperature and annual precipitation layers of 30 arc-seconds (~1 km) resolution

## Upper surface sculptures

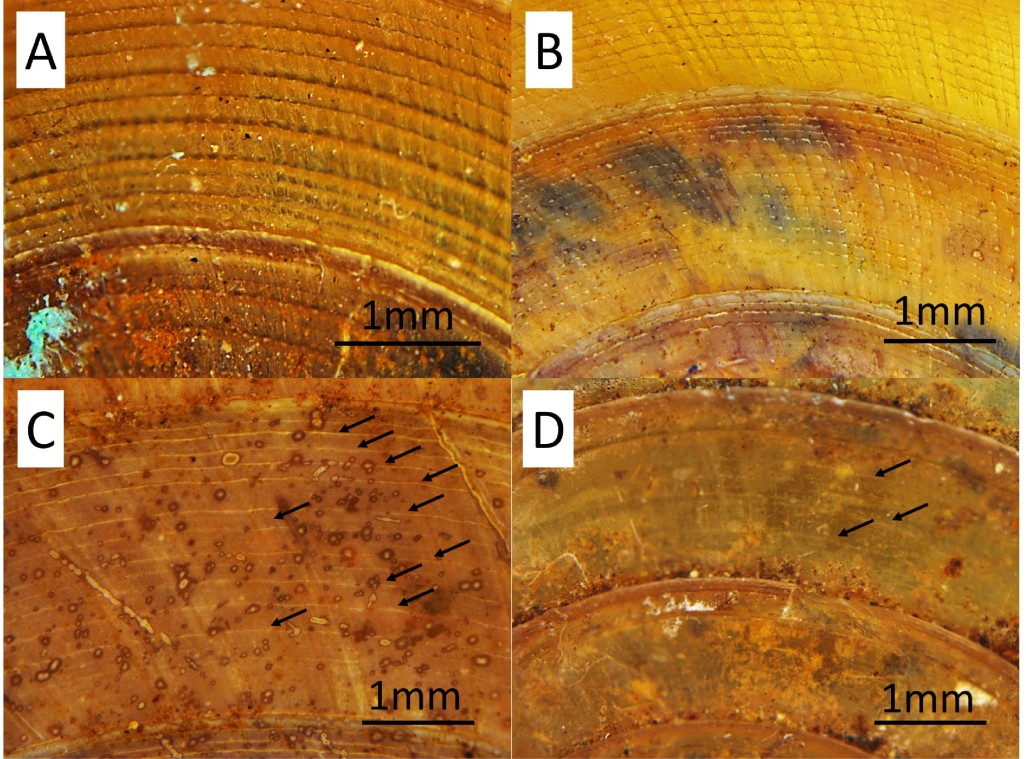

## Quantitative measurements

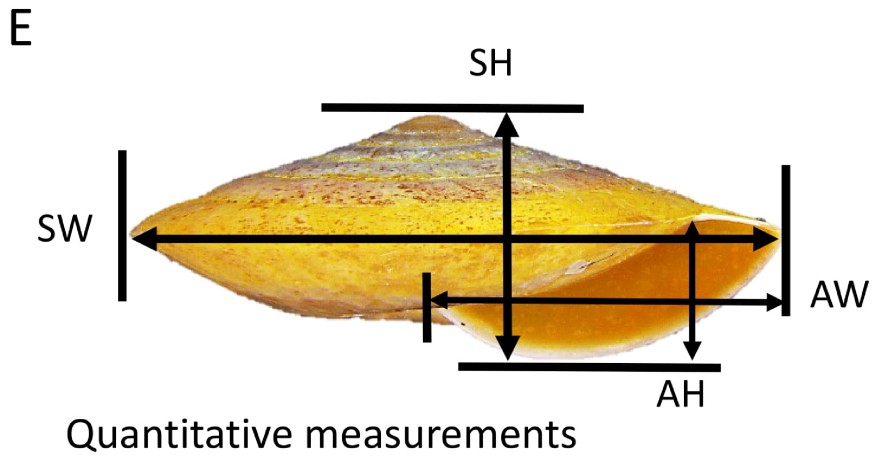

**Figure 2** **Upper surface sculptures and quantitative shell traits included in this study.** (A) Sculpture with spiral threads form nodes over radial sculpture (BOL/MOL 6312). (B) Sculpture with raised and distinct radial growth lines and thin spiral threads (BOL/MOL 6406). (C) Sculpture with indistinct radial growth lines and inconspicuous riblets and thin or very thin spiral threads (BOL/MOL 13061). (D) Sculpture with inconspicuous growth lines and low and thin spiral threads (BOL/MOL 890). (E) Four quantitative shell measurements: SH, Shell height; SW, Shell width; AH, Aperture height; and AW, Aperture width.

of WorldClim v1.4 database (http://www.worldclim.org) using point sampling tool of QGIS v2.60 (*QGIS Development Team, 2019*). As expected, the annual average temperature is confounding with the elevation. Hence, we explored the influence of the elevation and annual precipitation to the shell sizes and shell surface sculptures, as suggested by *Goodfriend (1986)*.

## Statistical analysis

For some species, the specimens are relatively uniform in shell upper surface sculpture and belong to one of the four categories of sculpture intensity. In contrast, other species are variable in shell upper surface sculpture and belong to more than one category (Table 1). Hence, we treated a specimen as an observation unit (i.e., replicates) for each of the four categories regardless of the specimen's species identity. We tested the null hypothesis ($H_0$) of there is no difference in the elevation of the habitat among the between the snail with different shell upper surface sculpture intensity. Besides, we also tested the null hypothesis ($H_0$) of there is no difference in the annual precipitation of the habitat among the between the snail with different shell upper surface sculpture intensity. As the data was not normally distributed, we performed Kruskal-Wallis tests to test the hypotheses (*Kruskal & Wallis, 1952*). Both analyses were performed in RStudio 1.1.4 (*RStudio Team, 2015*) (File S4).

We examined the collinearity of among the four shell size measurements. The results showed that aperture width (AW) is strongly correlated with shell width (SW) ($r = 0.99$), while the pairwise correlations among the other measurements are weaker with correlation coefficient values (r) range between 0.65 and 0.71. Hence, only SH, SW and AH measurements were retained for further analysis. All the three measurements were not normally distributed as reveal by Shapiro–Wilk test (*Shapiro & Wilk, 1965*). Therefore, Spearman's correlation tests (*Spearman, 1904*) were employed to examine the relationships between each of the two environmental variables with the three shell measurements.

## RESULTS

### Molecular phylogeny of *Trochomorpha* and *Geotrochus* species in Sabah

The final DNA alignment data matrix consists of 34 taxa and 1918 characters (16S: 1–461 bps; COI: 462–1,112; and ITS-1:1113–1918). The phylogenetic relationship of *Geotrochus* and *Trochomorpha* species of the concatenated dataset was shown in Fig. 3 (File S6). Generally, the phylogenetic trees estimated from each of the three genes show the topology as the tree estimated from the concatenated dataset (File S7). Generally, the three trees reconstructed based on the respective genes congruence to the tree that based combined genes, except for the taxa in Clade D. Particularly, *G. oedobasis*, *G. kitteli*, and *G. whiteheadi* that did not form a clade with *T. rhysa* in 16S and COI tree. On the other hand, all the taxa in Clade D appear to be polytomy in the ITS tree.

For concatenated DNA data matrix, the analyses of ML and BI yielded a phylogenetic tree with an identical topology that with >79% bootstrap values for ML and 1.00 posterior probability values for the four major clades. Both ML and BI analyses showed that *Geotrochus* and *Trochomorpha* species are not monophyletic. *Geotrochus kitteli* is the sister

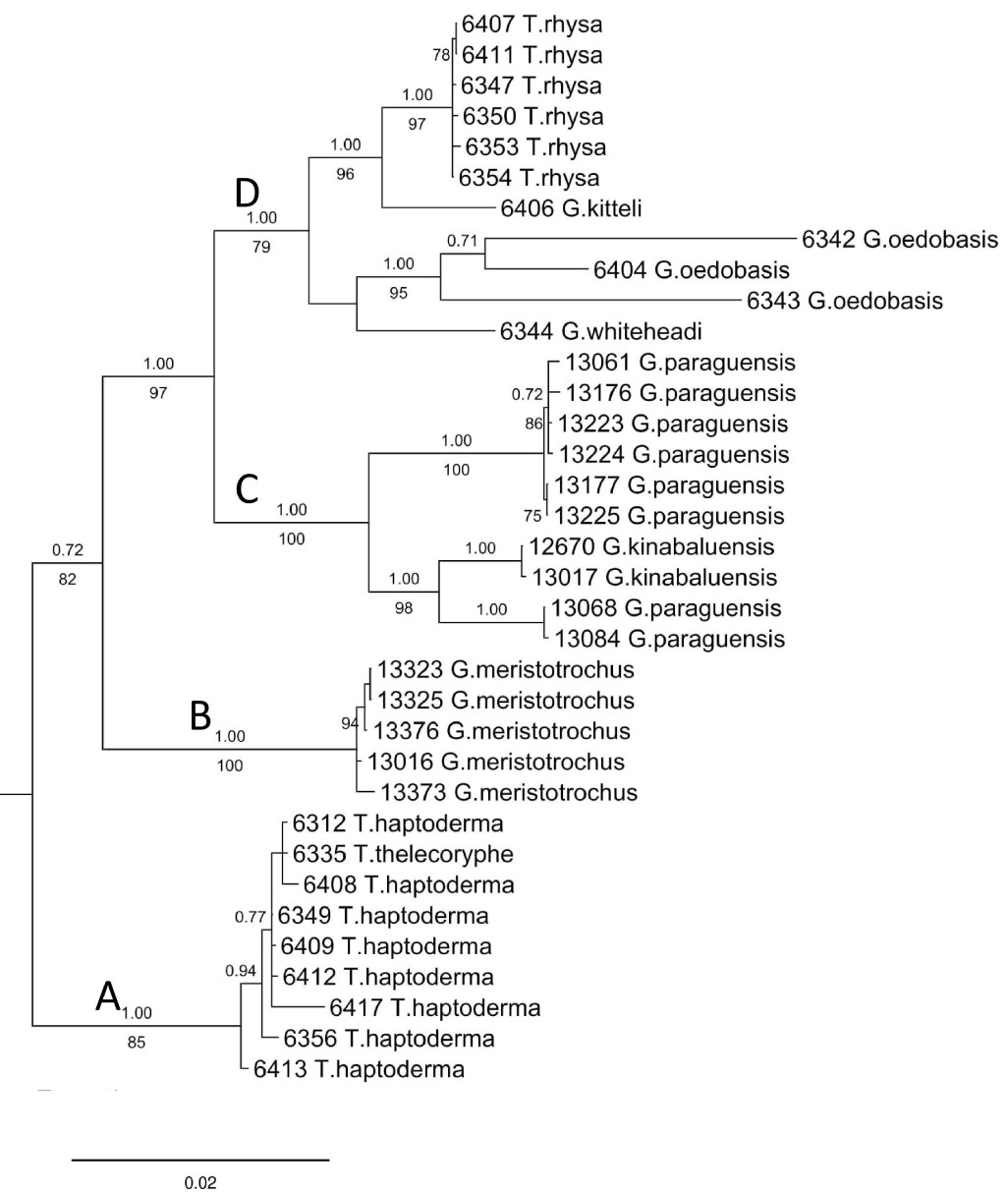

**Figure 3** **Bayesian inference tree of *Geotrochus* and *Trochomorpha* spp. based on concatenated dataset of 16S rDNA, COI and ITS-1 rooted to *Everettia klemmantanica*.** The letters A–D indicate the four major clades. Posterior probability (above the branch) from Bayesian inference and bootstrap support values (below the branch) from maximum likelihood analysis are indicated at the nodes with support values less than 0.7 of PP and 70% of BS were not shown in the figure. The number annotated in front of the species name was the BORNEENSIS collection number (Table 3).

taxon to *Trochomorpha rhysa* (Clade D), and *T. thelecoryphe* is nested in the *T. haptoderma* (Clade A). *Geotrochus paraguensis* from Banggi and Balambangan Island is paraphyletic with *G. kinabaluensis* (Clade C). Clade B contained *G. meristotrochus*.

**Table 3  Result of the phylogenetic signal test using Pagel's λ method and Blomberg's _K_ method.**

| Shell traits | Lambda (λ) | _p_-value | _K_ | _p_-value |
|---|---|---|---|---|
| Upper surface sculpture | 0.000 | 1 | 1.021 | 0.067 |
| Maximum shell height | 0.638 | 0.565 | 0.954 | 0.108 |
| Maximum shell width | 1.000 | 0.258 | 0.994 | 0.070 |
| Maximum aperture height | 0.000 | 1 | 0.700 | 0.339 |
| Maximum aperture width | 0.855 | 0.456 | 0.895 | 0.124 |

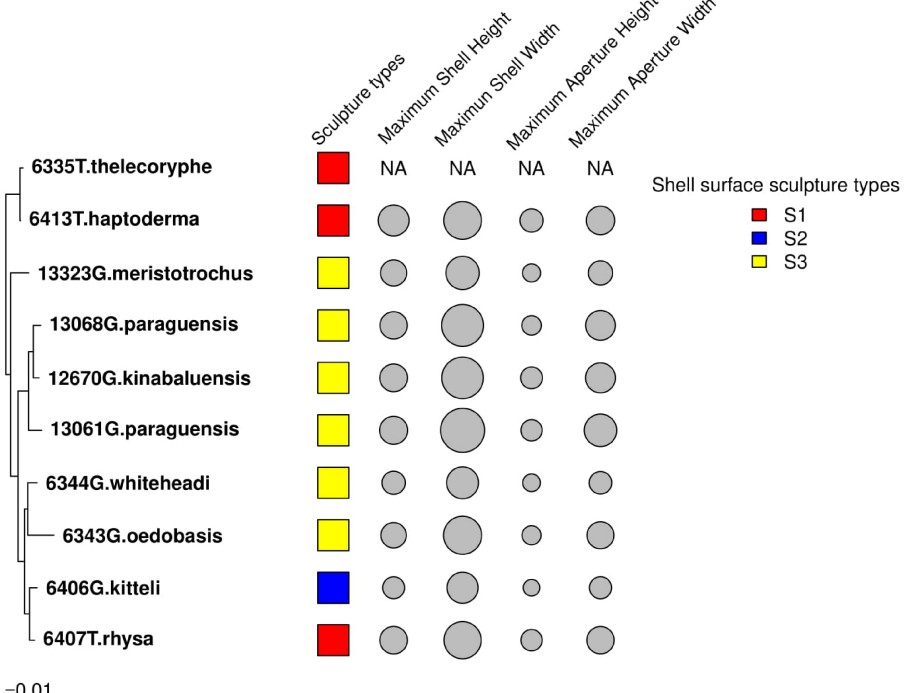

**Figure 4  Shell upper surface sculpture types and quantitative shell's traits were mapped on to the phylogenetic tree.** The shell upper surface sculpture types were represented by the different colour of the squares; and the four shell quantitative traits: maximum shell height, maximum shell width, maximum aperture height, maximum aperture width were represented by the size of the grey circle. The quantitative traits measurements were not available for _T. thelecoryphe_.

## Evidence for limited phylogenetic signal

The results from these two approaches showed that the shell height, shell width, aperture height and aperture width of _Geotrochus_ and _Trochomorpha_ considered in this study did not show significant phylogenetic signal. Besides, the shell upper surface sculptures appear as homoplasy character ($p > 0.05$) (Fig. 4 and Table 3).

## Association between shell morphology and environmental variables

The _Geotrochus_ and _Trochomorpha_ species that have coarser shell surface sculpture (i.e., Type S1 and S2) tend to occupy habitats at higher elevation (above 2000 m) (Kruskal-Wallis $X^2 = 118.36$, $df = 3$, $p < 0.0001$) and annual precipitation between 2400 mm and 2500

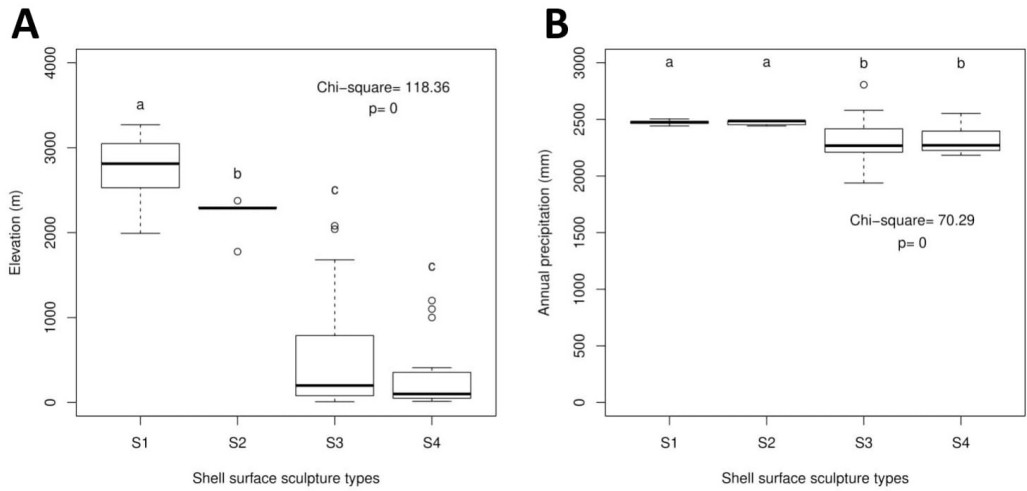

**Figure 5 Boxplots show the differences of the elevation and precipitation of the habitats of the shell with the four shell upper surface sculptures (S1–S4).** Kruskal-Wallis tests were performed, and the Chi-square values and the *p*-value of the test were shown in the plot. The alphabets above the boxplot indicate the results of multiple Wilcoxon signed-rank tests posthoc test. Sample sizes for each shell upper surface sculpture types were: S1 ($n = 77$); S2 ($n = 5$); S3 ($n = 58$); S4 ($n = 15$). (A) Differences of the elevation of the habitats of the shell with the four shell upper surface sculptures. (B) Differences of the annual precipitation of the habitats of the shell with the four shell upper surface sculptures.

mm (Kruskal-Wallis $X^2 = 70.29$, $df = 3$, $p < 0.0001$, Fig. 5, File S8). Shell width was negatively correlated with elevation ($r_s = -0.42$, $p < 0.0001$) and with annual precipitation ($r_s = -0.41$, $p < 0.0001$) (Fig. 6). On the other hand, shell height ($r_s = -0.14$, $p > 0.2$) and aperture height ($r_s = -0.02$, $p > 0.9$) were neither correlated with elevation nor annual precipitation ($r_s = -0.11$, $p > 0.3$; $r_s = -0.05$, $p > 0.6$).

# DISCUSSION

## Phylogeny of *Geotrochus* and *Trochomorpha* and its implication to taxonomy

The phylogenetic analysis showed that *Geotrochus* and *Trochomorpha* are not reciprocally monophyletic (Fig. 3). This result is contrary to the current taxonomy of the two genera that was based on the shell characters, especially the shell upper surface sculpture. The confusing taxonomy of the two genera goes back to the description of *Geotrochus* by *Van Hasselt* (1823, but published in *1824*) based on the specimens from Java Island, Indonesia, and the description of *Trochomorpha* by *Albers (1850)* based on several *Geotrochus*-like species from Southeast Asia and Pacific Islands. After that, *Von Martens (1867)* questioned the validity of the description of the genus *Geotrochus* by *Van Hasselt* (1823, published in *1824*) as there is no type assigned to the genus. Hence, *Von Martens (1867)* concluded that the *Geotrochus* is morphologically similar to *Trochomorpha*, and he used *Trochomorpha* instead of *Geotrochus* as a valid genus for the land snails from Borneo. Later, *Issel (1874)* used only *Trochomorpha* for the species recorded in Borneo with no mention of *Geotrochus* at all. Until the year 1935, *Pilsbry (1935)* validated the genus *Geotrochus* based on the

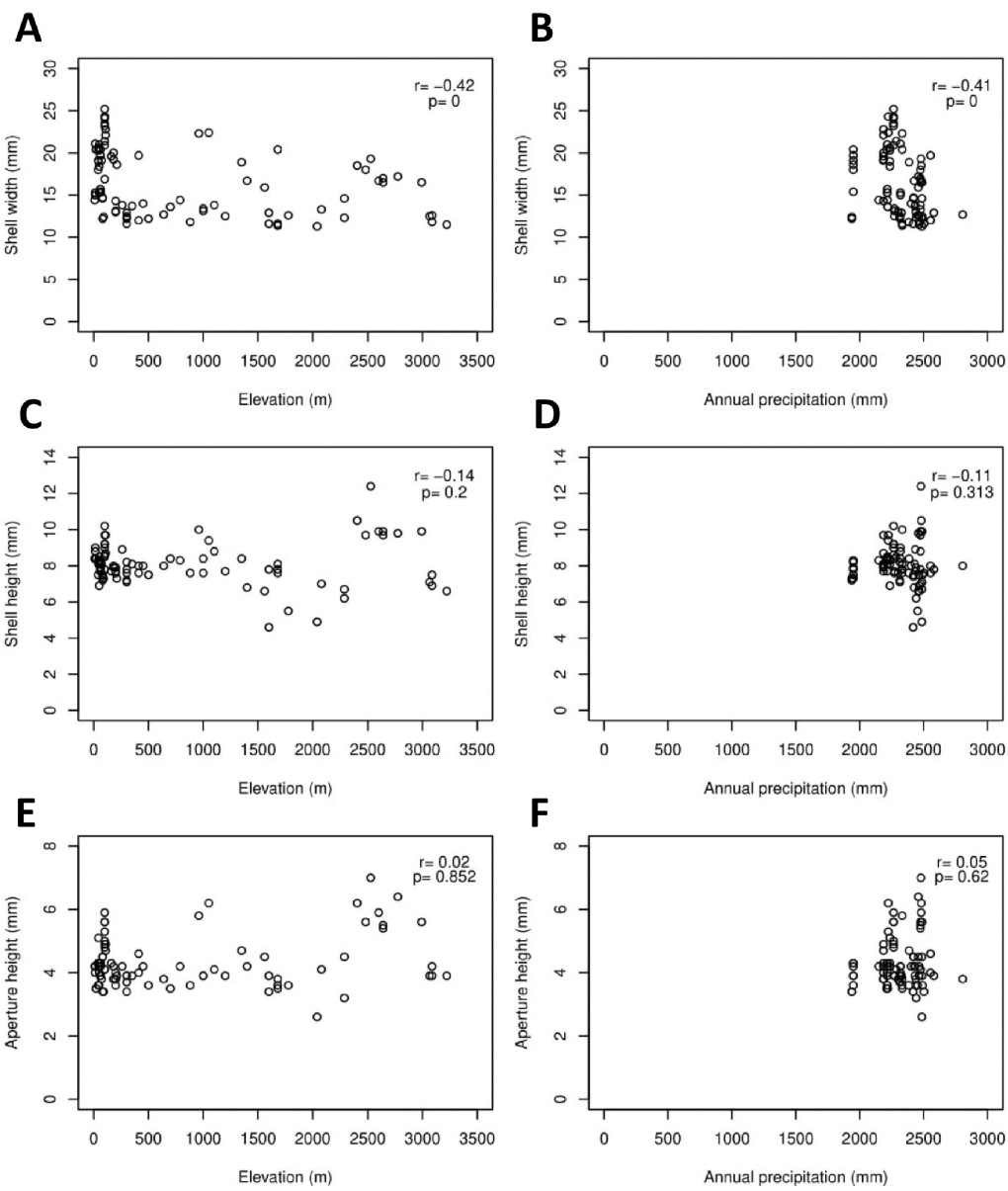

**Figure 6  Correlations between shell quantitative traits (i.e., sizes) and environmental variables (elevation and precipitation).** Spearman correlation tests were performed, and the correlation coefficient values ($r$) and the $p$-value of the test were shown in the plot, $n = 155$. (A) A significant negative correlation between shell width and elevation. (B) A significant negative correlation between shell width and annual precipitation. (C) No significant correlation between shell height and elevation. (D) No significant correlation between shell height and annual precipitation. (E) No significant correlation between aperture height and elevation. (F) No significant correlation between aperture height and annual precipitation.

Opinions no. 46 rendered by the International Commission on Zoological Nomenclature. *Solem (1964)* used only *Geotrochus* for the checklist of land snails in Sabah.

The first detailed shell and anatomical description of the species of the two genera in Sabah were done by *Tillier & Bouchet (1988)* based on *T. rhysa* from Mount Kinabalu.

Although the shell morphology, genitalia character and radula were described in detail, there was no comparison made to the known *Geotrochus* species or *Trochomorpha* species from other regions. In fact, *Geotrochus* was not mentioned at all in *Tillier & Bouchet (1988)*. The first comprehensive revision on *Geotrochus* and *Trochomorpha* is by *Vermeulen, Liew & Schilthuizen (2015)* for the species in Sabah based on the shell morphology. There were four *Trochomorpha* species, of which three were new, and 11 *Geotrochus*, of which six were new were included in the revision (*Vermeulen, Liew & Schilthuizen, 2015*).

The taxonomy history of the two genera in Sabah that lead to their confusing taxonomy is not merely an isolated case but reflects the taxonomy problem of the two genera on a large scale. The two genera have been used interchangeably as seen in the records of the two genera in the museum worldwide (File S1). As revealed by the GBIF data, there is a large extent of the overlapping in the distribution ranges of the two genera. This pattern could represent a real situation or could be resulted from the misidentification of the species or genera given the fact that the shells of the species in the two genera are very similar. *Schileyko (2002a)* and *Schileyko (2002b)* recognised current taxonomy of *Trochomorpha* is still unresolved, and he placed *Trochomorpha* in the Family Trochomorphidae whereas *Geotrochus* in the Family Helicarionidae.

Our results indicate that more comprehensive taxonomy study on *Trochomorpha* and *Geotrochus* are needed, not only for the Sabah taxa but for the entire distribution ranges of the two genera. However, the fact that *T. rhysa* is more genetically closely related to the *Geotrochus* species implies that its putative taxonomy position was likely misled by the parallelism in genital character as documented occasionally occurred in other groups of land snails (e.g., *Davison et al., 2005*; *Hirano, Kameda & Chiba, 2014*). Moreover, the coarse nodular upper surface sculpture was also taxonomically uninformative as the shell character has found evolved independently in this study.

Regarding taxonomy at the species level, this study confirmed the existence of the eight genetically distinct species classified by *Vermeulen, Liew & Schilthuizen (2015)*, except *T. thelecoryphe* and *T. haptoderma*. The two *Trochomorpha* species are very similar in the shell, but *T. thelecoryphe* has a flatter spire than *T. haptoderma* (*Vermeulen, Liew & Schilthuizen, 2015*). It is possible the type specimen of *T. thelecoryphe* in *Vermeulen, Liew & Schilthuizen (2015)* was a juvenile shell. Hence, more good condition specimens are needed for further clarification in a future study.

## Evolution of shell surface sculpture coarseness and shell sizes of *Geotrochus* and *Trochomorpha*

Polyphyly of the genus *Trochomorpha* indicated that the diagnostic shell upper surface sculpture is a homoplasy character. Our results show that environments of the habitat influence the shell characters, and phylogenetic closely related species do not tend to resemble each other in the shell size and shell upper surface sculpture. Hence, these shell traits of *Geotrochus* and *Trochomorpha* are evolutionary labile that are not suitable to be served as diagnostic characters at the genus level.

The convergence of the shell traits is instead a common phenomenon among land snails that occupying similar ecological niches (*Emberton, 1995*; *Phung, Heng & Liew, 2017*). The

physical shell is deemed to be the by-product of adaptation to their environmental attributes (*Goodfriend, 1986*; *Baur & Raboud, 1988*; *Pfenninger et al., 2005*; *Proćków, Kuźnik-Kowalska & Mackiewicz, 2017*; *Proćków et al., 2018*; but see *Gittenberger, 1991*; *Fehér et al., 2018* for non-adaptive radiation). The rough surface of the shell helps land snail live with excessive water or moisture in their habitats. For example, ribbed shells retain more water on the shell surface (*Giokas, Páll-Gergely & Mettouris, 2014*); hairy shells increase the snails' adherence wet surface of the plants in a more humid high-elevated area (*Pfenninger et al., 2005*; *Proćków et al., 2018*, but see *Shvydka, Kovalev & Gorb, 2019*); and coarser granular-like surface sculptures on shell help in reducing the water retention on the surface (*Nosonovsky & Bhushan, 2008*; *Maeda et al., 2019*). Besides, rough shells of other few ground-dwelling land snail species are known to be covered with soil that acts as camouflage (*Páll-Gergely et al., 2015*). From our field observation, we have not observed a shell of the living snail that is covered by soils or other materials. Hence, we suggest that the coarser shell surface helps *Trochomorpha* and *Geotrochus* species at highland elevation habitat dwell through fallen wet leaves by reducing the adhesiveness to its surrounding.

As can be seen from the records of the specimens and analysis (Table 1, Fig. 5), the snails with the coarser surface (i.e., S1 and S2) occur above 1500 m and more commonly above 2000 m. The abrupt transition of the shell surface is unlikely caused by temperature as the temperature generally decreases with increasing elevation (*Whitmore, 1975*; *Kitayama, 1992*; *Kitayama, 1994*). The occurrence of snails with a coarser shell upper surface (i.e., S1 and S2) at the area with relatively higher annual precipitation. Interestingly, these areas with relatively higher annual precipitation are also located at a higher elevation (>1,500 m).

In addition to rainfall, a substantial amount of precipitation may be added by horizontal rain in the cloud zone (*Kitayama, 1992*; *Kitayama, 1994*; *Kitayama et al., 1998*) or cloudy mossy forest (*Frahm et al., 1990*) between 2,000 m and 2,800 m. The habitat at this middle slope cloud zone with the increase in water surplus increased from 27% at 800m to 70% at 2,100 m (*Kitayama, 1994*; *Kitayama et al., 1998*). The species that predominantly with shell surface type S1 and S2 are endemic to Mount Kinabalu, namely, *T. haptoderma*, *T. rhysa*, *T. thelecoryphe*, and *G. kitteli* that are common above 2,000 m on the mountain.

The relationships between shell size and two significant environmental variables, namely, elevation and precipitation, are well documented (*Goodfriend, 1986*; *Baur & Raboud, 1988*; *Pfenninger & Magnin, 2001*; *Glass & Darby, 2009*; *Anderson, Lew & Peterson, 2003*; *Proćków, Kuźnik-Kowalska & Mackiewicz, 2017*). Our results show that the shell width and aperture width of the two genera are negatively correlated with elevation and annual precipitation. As the temperature is confounding with elevation, it also means that the shell size of the species in both genera follows converse Bergmann's rule (*Baur & Raboud, 1988*; *Anderson, Lew & Peterson, 2003*; *Proćków, Kuźnik-Kowalska & Mackiewicz, 2017*). It was hypothesised that the colder environment induces highland land snail to reach sexual maturity faster than those living in the warmer area. Hence, shells of the highland land snails are often smaller as the growth of the land snails is limited after maturity (*Proćków, Kuźnik-Kowalska & Mackiewicz, 2017*).

It is known that there is a positive relationship between high precipitation and shell size of land snails because humid habitat promotes the growth and expansion rate of shell whorls (*Goodfriend, 1986*). However, this may not be the case for montane species (*Goodfriend, 1986*; *Pročków, Kuźnik-Kowalska & Mackiewicz, 2017*). Our results show that *Geotrochus* and *Trochomorpha* species from sites with lower annual precipitation have a larger shell size. Although there is a statistically significant difference in the annual precipitation, we suggest that the precipitation *per se* might not be the only factor as the species of S1 and S2 that occur above 1500 m on the Mount Kinabalu are also experiencing horizontal precipitation resulted from the Middle slope wet cloud zone on Mount Kinabalu.

The negative correlation could probably due to the favourable effect of moisture on shell size has been compensated by the lower temperature on the high elevation that generally has a negative effect on shell size (*Goodfriend, 1986*; *Baur & Raboud, 1988*; *Anderson, Lew & Peterson, 2003*). Besides, decreasing in aperture size with the altitudinal gradient has generally been interpreted as an adaptation to the lower humidity at the lower elevational area (*Goodfriend, 1986*) as smaller apertures tend to lose proportionately more water per unit aperture area (*Goodfriend, 1986*).

## CONCLUSIONS

This study presents the first molecular phylogeny study on the genus *Geotrochus* and *Trochomorpha*. The phenotypically identified Sabah *Geotrochus* and *Trochomorpha* species do not congruent with the phylogenetic relationships. This incongruency is due to the homoplasy of upper surface sculpture which is used as the diagnostic character of the two genera. The coarser shell character may be an adaptation of the land snails to highland habitat with a more humid condition in the area. Besides, species at the lower elevation habitat tend to has a smaller shell. From the finding above, we concluded that the upper shell sculpture and shell size could not be used for the delimitation of Sabah *Geotrochus* and *Trochomorpha*. Hence, the current taxonomy of the two genera need further revision, and the future attempt should consider more samples that cover the entire distribution of the two genera.

## ACKNOWLEDGEMENTS

We thank Cornelius Peter for his assistance in molecular work. We also thank Edward Braun, Barna Páll-Gergely, Frank Koehler, and an anonymous reviewer for their constructive comments that improve this manuscript.

### Funding

This work was supported by the Ministry of Higher Education Malaysia via Universiti Malaysia Sabah under the research grant (GUG0187-2/2017; FRG0510). The funders had no role in study design, data collection and analysis, decision to publish, or preparation of the manuscript.

## Grant Disclosures

The following grant information was disclosed by the authors:
Universiti Malaysia Sabah: GUG0187-2/2017; FRG0510.

## Competing Interests

The authors declare there are no competing interests.

## Author Contributions

- Zi-Yuan Chang conceived and designed the experiments, performed the experiments, analyzed the data, prepared figures and/or tables, authored or reviewed drafts of the paper, and approved the final draft.
- Thor-Seng Liew conceived and designed the experiments, analyzed the data, prepared figures and/or tables, authored or reviewed drafts of the paper, and approved the final draft.

## Field Study Permissions

The following information was supplied relating to field study approvals (i.e., approving body and any reference numbers):

Field experiments were approved by the Sabah Parks for Mt. Kinabalu, Tambuyukon, Mahua, Banggi Island and Balambangan Island, and Yayasan Sabah for Inikea, Imbak Canyon and Maliau Basin (Permit: TTS/IP/100-6/2 Jld.7(70), 2018; Maliau Basin TTRP Project No. 228, 2017; and ICCA Expedition 2017).

## DNA Deposition

The following information was supplied regarding the deposition of DNA sequences:

Genbank, The COI, 16S and ITS-1 sequences here are accessible via GenBank accession numbers listed in Table 2.

COI sequences are accessible via GenBank: MK779460–MK779480 and MK811542–MK811552.

16S sequences are accessible via GenBank: MK334185–MK334205, MK335434, MK335445, MK335448–MK335450, MK335453, MK335461, and MK335462.

ITS sequences are accessible via GenBank: MK335433, MK335435–MK335444, MK335446, MK335447, MK335451, MK335452, MK335454–MK335460.

## Data Availability

Raw data, Trochomorpha and Geotrochus records, ModelTest results, sequences, and R script are available in the Supplemental Files.

## Supplemental Information

Supplemental information for this article can be found online at http://dx.doi.org/10.7717/peerj.10526#supplemental-information.

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
