# Peer review of "A molecular phylogeny of Geotrochus and Trochomorpha species (Gastropoda: Trochomorphidae) in Sabah, Malaysia reveals convergent evolution of shell morphology driven by environmental influences"

_PeerJ, doi:10.7717/peerj.10526_

## Round 0.1 · original submission · Major Revisions

Your manuscript has been reviewed by three experts and I am prepared to make a decision. As you can see, the reviews differed in their views on the manuscript. Ultimately, I feel your manuscript may be suitable after a major revision.

I would like you to pay close attention to the reviews and revise the manuscript take as much of their advice as possible.

Reviewer 2 provided a fairly negative review and one of the reviewer's concerns was the uneven number of specimens per species. I spent a long time thinking about this issue (so much so that I have to offer an apology for letting this sit on my desk). Although I share the reviewer's concern I ultimately decided that your manuscript contains valuable information.

The limited phylogenetic signal in the morphological data could actually an advantage. The information in your Table 5 and (especially) Figure 7 is suggests that it might be reasonable to view the quantitative traits you examined as coming from a single population (obviously, the shell sculpture has strong differences among taxa, but the graphics in Figure 7 show little variation among taxa). Thus, the simple correlations seem reasonable to me.

I would like the authors to rearrange their results and describe the phylogenetic signal *before* their environmental correlations. This would make the order of their results:

1. Molecular phylogeny of Trochomorpha and Geotrochus species in Sabah
2. Phylogenetic signal
3. Association between shell morphology and environmental variables

(note that I would change the title of "Phylogenetic signal" to "Evidence for limited phylogenetic signal". Also the statement that "...all diagnostic shell characters of Geotrochus and Trochomorpha considered in this study exhibited a weak phylogenetic signal" doesn't really make sense to me. My reading of the results is that the quantitative traits are all over the place whereas the sculpture patterns appear homoplastic.

I am not concerned regarding the criticism of Pagel's lambda. No method is perfect and the idea behind lambda (multiplying internal branches by lambda and then fitting Brownian motion) can certainly be criticized, but it is not unreasonable in my opinion.

A second concern I have is that incomplete lineage sorting could have an impact on the conclusions. There is no way to get at this issue in a really definitive manner unless phylogenomic data are available. However, the authors have two loci (COI and parts of the ribosomal repeat). They should present three trees - the combined tree (which they already present), a COI tree, and a 16S+ITS tree. The partitioning the author used is fine.

If the trees are very different there are two potential reasons: 1) low power of one or both of the individual trees (the mitochondrial COI tree and the ribosomal tree); or 2) there is genuine discordance among the gene trees. If support for one of the two trees is generally low it would favor the first possibility (low power) whereas high support would favor genuine discordance.

One other issue is the use of GAMMAI in RAxML. Although GAMMAI may be reasonable, there are concerns about the interaction between the gamma shape parameter and the proportion of invariant sites. I recommend the authors simply use IQ-TREE and allow IQ-TREE to do the model selection for each partition.

Finally, the authors provide their sequence data as a fasta file. They should provide the partition boundaries. They could upload the nexus file for MrBayes and/or the files used for IQ-TREE/RAxML (relaxed phylip and a partition file). This will make their results more reproducible.

I hope this guidance is helpful and that the authors are doing well during these trying times. I look forward to the submission of a revised manuscript.

·

Basic reporting

English is good, but should be finetuned.
Literature should be a bit extended, see in my general remarks.

Experimental design

see my comments in General comments for the author.

Validity of the findings

see my comments in General comments for the author.
Conclusions: Yes, conclusions are fine, such as a genus is not monophyletic, or a species is nested in another species. However, I, as a taxonomist, don't like to stop here, and strongly suggest to transfer your phylogenetic data into taxonomic changes.

Discussion could be extended a bit with what I suggested.

Additional comments

This is a nice study, definitely worth publishing, however, there are certain problems I can immediately see and I report below.

(1) Molecular phylogeny: Before you analyse the entire matrix, you should produce separate mitochondrial and nuclear trees, and check whether their topology matches. I may have been overlooked, but I could not find any traces of this. Mitochondrial and nuclear trees should be made separately available, even if as supplementary material.

(2) While you present a useful and important phylogeny, I was disappointed to see that unfortunately this is again a study with good phylogenetic information, but insufficient morphological/taxonomic conclusions.

First, the type species of Geotrochus van Hasselt, 1823 is Helix conus L. Pfeiffer, 1841 from Java. The type species of Trochomorpha Albers, 1850 is Helix trochiformis L. Pfeiffer, 1842 (complicated situation, H.B. Baker, 1941 established Trochomorpha typus [from the Society Islands] as a new species, and subjective synonym of Helix trochiformis Pfeiffer, 1842).

You cannot talk about genera without examining the type species. Genera are based on type species. Both type species are probably available in museums, at least you should examine and publish photographs, and arrive to a conclusion based on morphology without molecular phylogeny. I already know what answer I will receive (this is beyond the scope of the present paper, etc.), but please try not to stop at stages like "this and this genus is not monophyletic". With these kind of publications you just increase the phylogeny-taxonomy gap (Franz 2005).

It is OK that you did not include those species in your phylogentic analysis, but I am sure nobody will be able to perform a phylogenetic study with those two type species in the next 50 years or so. So you would need some explanation on the morphology and arrive to a conclusion. I would strongly suggest to synonymize these two genera, or state that you suggest using Geotrochus because its types species occurs in the same geographic area, or whatever, so at least until more information becomes available, make some order and get rid of one of the generic names that causes troubles.

Also, the type locality of the type species of Trochomorpha is not on the map. I understand that that record might not have contained in the database you used, but that means your database is wrong, and you are strongly suggested to make a map with reliable localities.

Second, T. thelecoryphe and T. haptoderma are almost certainly synonyms. I don't see why you don't synonymize them in this paper. Both species were described recently by you!

(3) "The physical shell is deemed to be the by-product of adaptation to their environmental attributes": Well, if you want to support this view, you can find citations for sure. However, there are papers proving that shell morphology is often a result of random genetic drift, and the traits are not to be automatically explained as adaptation to the environment (e.g. Fehér et al. 2018).

(4) Role of rough upper (dorsal) surface: In Plectopylidae, most living species have reticulated sculpture on the dorsal shell side, which is often covered with soil and this may be of value in providing camouflage (Páll-Gergely et al. 2015). This paper should be cited. Are trochomorphid specimens clean or dirty in the wild?

(5) I am not good at statistical analysis, but Fig. 4 about precipitation does not seem to be convincing. The mean for S1 and S2 is roughly 2500 mm, whereas that of S3 and S4 is 2300 m with huge interval. This 200 mm differences is nothing, and can easily be sampling error (in other words you cannot have precise enough data on precipitation for these kind of studies).

References

Fehér, Z., Jaksch, K., Szekeres, M., Haring, E., Bamberger, S., Páll-Gergely, B., Sattmann, H. & Sólymos, P. (2018): Range-constrained co-occurrence simulation reveals little niche partitioning among rock-dwelling Montenegrina land snails (Gastropoda: Clausiliidae). — Journal of Biogeography 45: 1444–1457.
Franz, N.M (2005): On the lack of good scientific reasons for the growing phylogeny⁄classification gap. Cladistics 21: 495–500.
Páll-Gergely B, Hunyadi A, Ablett J, Lương HV, Naggs F, Asami T (2015) Systematics of the family Plectopylidae in Vietnam with additional information on Chinese taxa (Gastropoda, Pulmonata, Stylommatophora). ZooKeys 473: 1–118.

Reviewer 2 ·

Basic reporting

Main concerns:

The fact that the geographical scope of the study is restricted to one

single state of Malaysia (northern tip of Borneo Island) and not the whole

range of the genera in question makes the final conclusions much weaker.

Furthermore, conclusions are even weaker if not all Sabah species is

involved in the study.

my largest concern about this study that authors claim to 'demonstrate' a

relationship between shell traits and environmental factors. This should

be consulted with someone who is more versed in statistics. But I suspect

that it is not OK that the number of elements per species are different.

If we are curious about intra-specific issues, several specimens of ONE

species should be compared. If we are curious about the genus-group level,

we should include ONE specimen per each species (something suggested by

Fig 7). Maybe one can argue that it makes sense to use one specimen per

POPULATION, so species of larger area are better represented. But using

ALL available specimens when species are unequally represented in the

sample - that seems problematic to me. But, again I think a statistics

expert could better tell what is the problem with it.

some other points:


measured dimensions. Only shell and aperture width values are measured, but based on the pictures these species can be distinguished by many other shell features, e.g. the convexity of the cone, shape of shell, the strength of the keel, etc. these traits are all ignored by the stat analysis.


Discussion

Lines 189-190: Geotrochus and Trochomorpha are not reciprocal
monophyly --> not reciprocally monophyletic

Lines 190-191: "This result is contrary to the current taxonomy of the two

genera that based on the shell characters, especially the shell upper

surface sculpture" This sentence should be the most important conclusion

of this study. i.e. morphology-based current system is not in consensus

with the phylogenetic tree. I agree. However, see my note about the phylo

signal and table 5. Sculpture is a trait where Blomberg K indicates

phylogenetic signal!!! So how can you interpret this???

Line 205: I do not clearly understand what the authors wanted to mean

here...

Lines 211-213 If I understand well, here the authors call a species "new"

if it was described in 2015. They are obviously not "old" species, but in

the literature a taxon is called "new" if it is described in the same

publication.


Conclusions:

I am not convinced that "...The coarser shell character
290 may be an adaption of the land snails to highland habitat with a more

humid condition in the area" I still believe that excluding the role of

pure chance needs stronger statistical evaluation.

"...the current taxonomy of the two genera
294 need further revision and the future attempt should consider more

samples that cover the entire
295 distribution of the two genera"

I strongly lack any kind of taxonomical/nomenclatural conclusion at the

end. I feel that the statistical support for the relationship between

environmental factors and shell traits is weak, so this remains the only

possible conclusion. If a tree is non-monophyletic it can primarily due to

(i) bad taxonomy (ii) problems with the chosen markers in tree

reconstruction (see the review of Funk and Omland 2003). If I am right,

authors vote here for the first case. If so, they should make a

conclusion. Otherwise, why did they make this study? A conclusion like

"...need further investigation" I feel too little for a renowned journal

like PeerJ.

Lines 151-153: I think a species can have a sister species and a family

can have a sister family... a species can hardly be a sister of a whole

family unless it is a monospecific family

Lines 183-185
other K's are closer to 1 than 0, although p's are poor. But this might be

due to different other factors, like the number of studied specimens not

only the lack of signal. And also some lambdas are pretty high (though

p's are poor).
Another point, several experts argue that these metrics for phylosignal are

quite outdated, see e.g. https://www.carlboettiger.info/2013/10/11/is-it-

time-to-retire-pagels-lambda.html

Table 5. Contrary to the authors' claim, the sculpture's K value seems

significant. Some other lambdas and Ks are also quite high (although p-

values are non-significant). But being unable to demonstrate the presence

of phylogenetic signal does not necessarily equal to the absence of the

phylogen signal. Might also be due to low N, or generally due to the fact

that many authors believe that Pagel's test is really outdated.



Figure 7 caption: "Figure 6 Visualization...." this was probably

overlooked. But more important points here: what about S4 type?? that is

not illustrated. in contrast with Fig 4, where there is S4.
Why G. paraguensis is represented by two entities when others only by one?

does it mean that N=10 was the number of elements in the statistical test

(and not 155)?

Experimental design

see above

Validity of the findings

see above

·

Basic reporting

The manuscript is brief and concise in style, the language used, although grammatically not always perfect, is sufficiently clear and understandable. Language editing could help getting rid of some linguistic imperfections, but I don't accept this as my responsibility.

Experimental design

The problem is clearly described, relevant literature is considered, the methods and statistical tests are suitable to address the problem. The have been executed properly.
All relevant data is made available.

Validity of the findings

The conclusions of the paper are reasonable and supported by the evidence presented.

Additional comments

Nice work. With a little bit of linguistic polishing, I am sure that this has the potential to become a good example study demonstrating that sole reliance of shell features can be problematic in land snail systematics.

I suggest minor revision to give the authors the opportunity to work on the linguistic aspects of presenting their work.

---

## Round 0.2 · accepted · Accept

The new rounds of reviews were extremely positive. I agree that you have addressed all of the reviews and have produced an excellent paper.

I am delighted to accept your manuscript and hope you accept my apologies for the delay in this decision. It was entirely my mistake and does not reflect a delay on the part of the reviewers.

·

Basic reporting

I have read the explanations of the authors, and I accept them. I still "like" if studies like this have stronger phylogenetic/taxonomic base, but I understand that in the present case the scientific question is not taxonomic.

Experimental design

I am satisfied with the explanations of the authors.

Validity of the findings

I am satisfied with the responses of the authors.

Reviewer 2 ·

Basic reporting

a short note on literature references. Some new papers were added to the revised text. I have found that Pall-Gergely et al 2015 is, for example, mentioned in the text but not listed among references. Please double-check the whole text. Also, keep an eye on the alphabetic order of References

Experimental design

no comment

Validity of the findings

see comments below

Additional comments

I still have some of the concerns written in my first review, However, it has to be admitted that authors improved the ms significantly and gave (or in some cases made a significant effort to give) answers to all the points. So this improved version seems worthy of publication